# SPAG7 deletion causes intrauterine growth restriction, resulting in adulthood obesity and metabolic dysfunction

Stephen E Flaherty III[1], Olivier Bezy[1], Brianna LaCarubba Paulhus[1], LouJin Song[1], Mary Piper[1], Jincheng Pang[1], Yoson Park[1], Shoh Asano[1], Yu-Chin Lien[2,3], John D Griffin[1], Andrew Robertson[4], Alan Opsahl[4], Dinesh Hirenallur Shanthappa[1], Youngwook Ahn[5], Evanthia Pashos[1], Rebecca A Simmons[2,3], Morris J Birnbaum[1], Zhidan Wu[1]*

[1]Internal Medicine Research Unit, Pfizer Inc, Cambridge, United States; [2]Center for Research on Reproduction and Women's Health, Perelman School of Medicine, University of Pennsylvania, Philadelphia, United States; [3]Division of Neonatology, Department of Pediatrics, Children's Hospital of Philadelphia, Philadelphia, United States; [4]Drug Safety Research and Development, Pfizer Inc, Groton, United States; [5]Medicine Design, Pfizer Inc, Cambridge, United States

**Abstract** From a forward mutagenetic screen to discover mutations associated with obesity, we identified mutations in the *Spag7* gene linked to metabolic dysfunction in mice. Here, we show that SPAG7 KO mice are born smaller and develop obesity and glucose intolerance in adulthood. This obesity does not stem from hyperphagia, but a decrease in energy expenditure. The KO animals also display reduced exercise tolerance and muscle function due to impaired mitochondrial function. Furthermore, SPAG7-deficiency in developing embryos leads to intrauterine growth restriction, brought on by placental insufficiency, likely due to abnormal development of the placental junctional zone. This insufficiency leads to loss of SPAG7-deficient fetuses in utero and reduced birth weights of those that survive. We hypothesize that a 'thrifty phenotype' is ingrained in SPAG7 KO animals during development that leads to adult obesity. Collectively, these results indicate that SPAG7 is essential for embryonic development and energy homeostasis later in life.

*For correspondence:
wu_zhidan@hotmail.com

## eLife assessment

This study combines molecular genetics and target validation to discover genes involved in obesity and determine their role. It was unanimously agreed that the work is **important** in terms of significance as it has conceptual and practical implications beyond metabolism, including embryonic and placental development. The strength of evidence is **convincing** from the use of their forward genetic screen in mice.

## Introduction

Obesity and associated metabolic disorders, such as insulin resistance and Type 2 Diabetes (T2D), are major sources of morbidity and mortality that are reaching epidemic proportions (*Cawley et al., 2021*; *Kelly et al., 2008*; *Stokes and Preston, 2017*; *WHO, 2016*; *WHO, 2020*; *Zimmet et al., 2016*). Lifestyle interventions, such as diet and exercise, can be effective in combating these afflictions in some patients. However, low adherence and counter-regulatory mechanisms that limit weight reduction and long-term weight maintenance impede the effectiveness of such interventions (*Brownell, 1998*; *Miller et al., 2002*;

**eLife digest** Obesity rates are climbing worldwide, leading to an increase in associated conditions such as type 2 diabetes. While new pharmaceutical approaches are available to help individuals manage their weight, many patients do not respond to them or experience prohibitive side effects. Identifying alternative treatments will likely require pinpointing the genes and molecular actors involved in the biological processes that control weight regulation.

Previous research suggests that a protein known as SPAG7 could help shape how mice use and store the energy they extract from food. Flaherty et al. therefore set out to investigate the role this protein plays in the body. To do so, they created a line of mice born without SPAG7, which they monitored closely throughout life.

These animals were underweight at birth and did not eat more than other mice, yet they were obese as adults. Their ability to exercise was reduced, their muscles were weaker and contained fibers with functional defects. The mice also exhibited biological changes associated with the onset of diabetes. Yet deleting SPAG7 during adulthood led to no such changes; these mice maintained normal muscle function and body weight.

Closely examining how SPAG7-deficient mice developed in the womb revealed placental defects which likely caused these animals to receive fewer nutrients from their mother. Such early-life deprivation is known to be associated with the body shifting towards maximizing its use of resources and privileging fat storage, even into and throughout adulthood.

By shedding light on the biological role of SPAG7, the work by Flaherty et al. helps to better understand how developmental events can increase the likelihood of obesity later in life. Further investigations are now needed to explore whether this knowledge could help design interventions relevant to human health.

*Roberts and Barnard, 2005*). The new anti-obesity drugs, semaglutide, a GLP-1 receptor agonist, and tirzepatide, a dual GIP and GLP-1 receptor agonist, have demonstrated remarkable efficacy for weight loss and hold great promise as anti-obesity treatments to alleviate metabolic dysfunction (*Venniyoor, 2022*; *Wilding et al., 2021*). However, even with their recent success, there are many patients who do not respond to GLP-1 receptor agonist treatment or have to discontinue the treatment due to side effects (*Chao et al., 2022*). Thus, there remains a need for additional new therapies to combat obesity.

To identify new targets for the treatment of obesity and diabetes, we began a large-scale forward genetic phenotypic screen in mice (*Wang et al., 2015*). Random mutations in the mouse genome were generated using the chemical ENU (N-ethyl-N-nitrosourea). The genomes of the animals were sequenced, and various metabolic endpoints were evaluated to identify genes with mutations associated with obesity and insulin resistance phenotypes (*Figure 1A*). Through this process, mutations in the gene *Spag7* (sperm-associated antigen 7) were identified. SPAG7 is well-conserved with 97% of the amino acid sequence being identical in humans and mice. Structurally, it contains a nuclear localization signal, making it likely a nuclear protein, and an R3H domain (*Nagata et al., 2005*). The function of the R3H domain is also not fully understood; however, it is predicted to bind polynucleotides, and other proteins that contain R3H domains are known to bind single-stranded DNA or RNA (*Grishin, 1998*; *He et al., 2013*).

The biological function of SPAG7 has been largely unstudied. It was first identified in the inner acrosomal compartment of fox sperm (*Beaton et al., 1994*). However, the gene is expressed in every tissue and cell in humans and mice (*Noguchi et al., 2017*). It has been implicated in various disease states, including Asperger syndrome (*Tentler et al., 2003*), parvovirus B19 infection (*Kerr et al., 2005*), synovial sarcoma (*Fernebro et al., 2006*), azoospermia (*Ayhan et al., 2014*), periodic fever, aphthous stomatitis, pharyngitis, and adenopathy (PFAPA) syndrome (*Bens et al., 2014*), squamous cell carcinoma (*Singh et al., 2020*), and oligoasthenozoospermia (*Abu-Halima et al., 2023*). However, the role SPAG7 plays in these diseases remains elusive.

Mice carrying SPAG7 mutations displayed interesting metabolic phenotypes, suggesting that SPAG7 may play a role in regulating metabolism in vivo. The goals of this study are to determine whether SPAG7 loss of function affects systemic metabolism and to establish a mechanism of action for SPAG7.

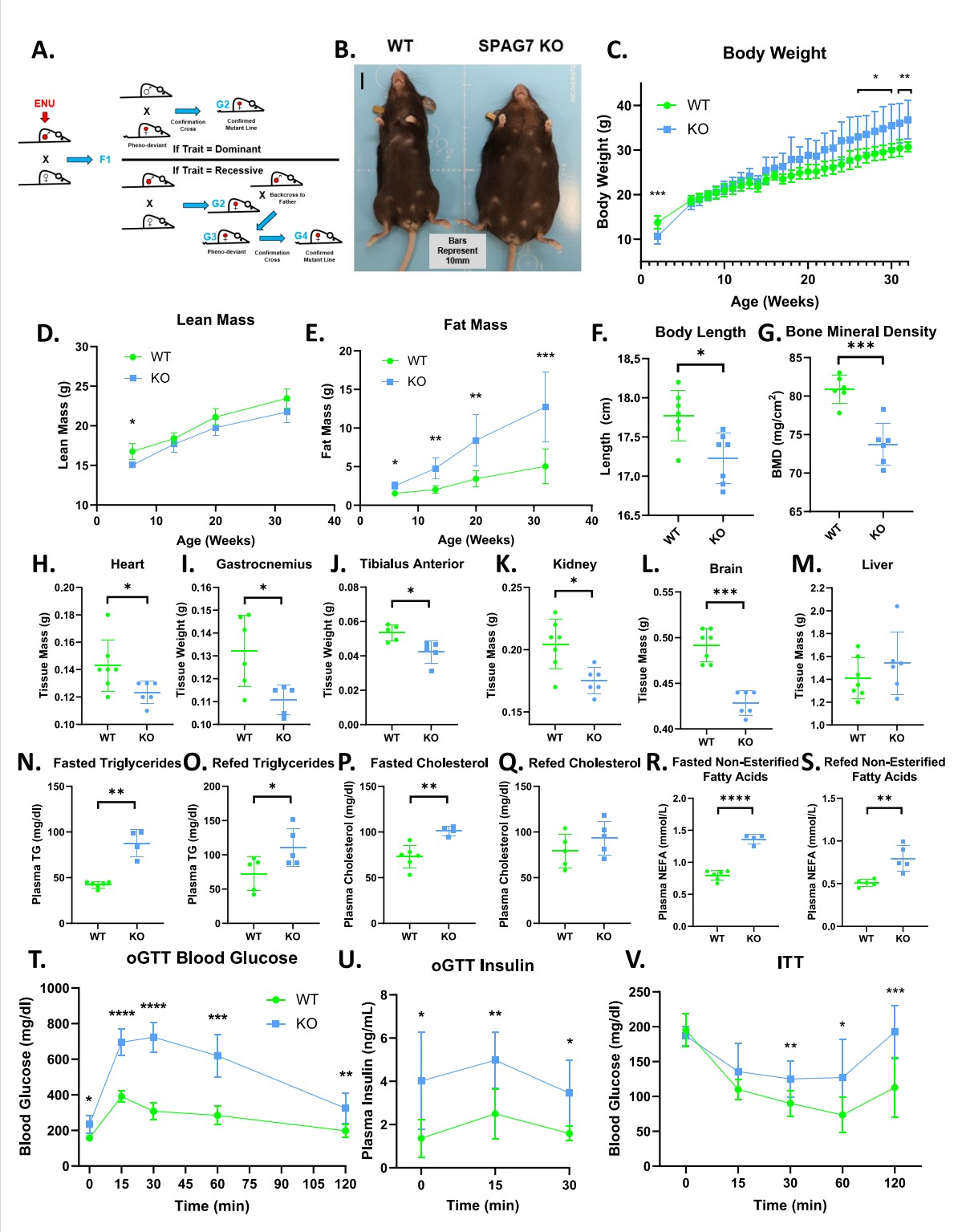

**Figure 1.** SPAG7-deficiency causes obesity and insulin resistance. (**A**) Graphical representation of an ENU-driven forward genetic screen. (**B**) Images of WT and SPAG7 KO littermates at 32 weeks of age. (**C**) WT vs SPAG7 KO body weight over time. n=7. Significance was assessed by Welch's two sample t-test. (**D**) WT vs SPAG7 KO lean mass over time. n=7. Significance was assessed by Welch's two sample t-test. (**E**) WT vs SPAG7 KO fat mass over time. n=7. Significance was assessed by Welch's two sample t-test. (**F**) Body length measured from nose to base-of-tail. n=7. Significance was assessed by Welch's two sample t-test. (**G**) Bone mineral density, as determined by DEXA scan. n=7. Significance was assessed by Welch's two sample t-test. (**H**) Heart weight. n=7. Significance was assessed by Welch's two sample t-test. (**I**) Gastrocnemius muscle weight. n=7. Significance was assessed

*Figure 1 continued on next page*

*Figure 1 continued*

by Welch's two sample t-test. (**J**) Tibialis anterior muscle weight. n=7. Significance was assessed by Welch's two sample t-test. (**K**) Kidney weight. n=7. Significance was assessed by Welch's two sample t-test. (**L**) Brain weight. n=7. Significance was assessed by Welch's two sample t-test. (**M**) Liver weight. N=7. (**N**) Plasma triglyceride levels following 8 hr fast. n=5. Significance was assessed by Welch's two sample t-test. (**O**) Plasma triglyceride levels following 8 hr fast with a 2 hr ad-lib refeed. n=5. Significance was assessed by Welch's two sample t-test. (**P**) Plasma total cholesterol levels following 8 hr fast. n=5. Significance was assessed by Welch's two sample t-test. (**Q**) Plasma total cholesterol levels following 8 hr fast with a 2 hr refeed. n=5. (**R**) Plasma NEFA levels following 8 hr fast. n=5. Significance was assessed by Welch's two sample t-test. (**S**) Plasma NEFA levels following 8 hr fast with a 2 hr refeed. n=5. Significance was assessed by Welch's two sample t-test. (**T**) Blood glucose levels following an oral glucose challenge. n=7. Significance was assessed by Welch's two sample t-test. (**U**) Plasma insulin levels following an oral glucose challenge. n=7. Significance was assessed by Welch's two sample t-test. (**V**) Blood glucose levels following an IP insulin challenge. n=7. Significance was assessed by Welch's two sample t-test. * $p<0.05$, ** $p<0.01$, *** $p<0.001$, **** $p<0.0001$.

The online version of this article includes the following figure supplement(s) for figure 1:

**Figure supplement 1.** SPAG7 KO adipose tissue and liver.

## Results

### SPAG7-deficient animals are obese and insulin resistant

To investigate the role of SPAG7 in metabolism, we generated a germline SPAG7-deficient mouse model (SPAG7 KO). SPAG7 KO animals are obese in adulthood (*Figure 1B*). But they have an unexpected growth curve; they are underweight early in life, catch up around six weeks of age, and then continue to outgrow WT littermates into and throughout adulthood (*Figure 1C*). SPAG7-deficient mice demonstrate decreased lean mass early in life that remains below that of WT littermates throughout their lifespan (*Figure 1D*). In contrast, fat mass in the animals increases continually, growing to three-to-five-times the fat mass of WTs (*Figure 1E*). SPAG7-deficient animals display decreased body length, measured nose to base of tail (*Figure 1F*). Bone mass density is also decreased in SPAG7 KO animals, which may contribute to the decreased lean mass (*Figure 1G*).

The majority of the increased body weight in SPAG7-deficient animals stems from increased adipose tissue mass (*Figure 1—figure supplement 1A–D*). All adipose tissue depots more than double in size (*Figure 1—figure supplement 1A–D*). SPAG7-deficient adipose tissues display adipocyte hypertrophy and increased immune cell infiltration, consistent with chronic obesity (*Figure 1—figure supplement 1E–H*). However, heart, skeletal muscle, kidney, and brain weights are all decreased in SPAG7-deficient animals (*Figure 1H–L*). Livers display no change in mass but do have increased triglyceride content, indicating hepatic steatosis (*Figure 1M*, *Figure 1—figure supplement 1I–K*). No changes in circulating liver enzymes were observed, indicating that the lipid accumulation was not severe enough to induce significant liver damage (*Figure 1—figure supplement 1L–O*). Plasma cholesterol, triglycerides, and free fatty acids are all increased, consistent with an obesity phenotype (*Figure 1N-S*).

SPAG7-deficient animals display severely impaired glucose uptake in response to oral glucose gavage (*Figure 1T*). Glucose-stimulated insulin secretion is increased in these animals, consistent with an insulin resistance phenotype (*Figure 1U*). The animals also display an insulin intolerance phenotype, with decreased glucose uptake following an insulin injection (*Figure 1V*). This is, perhaps, unsurprising, given the rather extreme fat mass phenotype.

### Obesity in SPAG7-deficient animals is caused by decreased energy expenditure

A change in food intake was not detectable in SPAG7-deficient animals (*Figure 2A*). In fact, a trend towards decreased food intake was consistently observed. When placed in metabolic cages, SPAG7-deficient animals display a significant decrease in energy expenditure that is present during the light cycle, but particularly clear in the dark cycle, when we would expect the animals to be active and moving about the cage (*Figure 2B and C*). Consistent with this finding, SPAG7-deficient animals display decreased locomotor activity, as measured by home-cage beam break (*Figure 2D*).

To determine if thermogenesis also contributes to the decreased energy expenditure observed in the SPAG7 KO mice, SPAG7-deficient animals were housed at thermoneutrality (TN). At TN the animals display the same phenotypes: increased body weight in adulthood, dramatically increased fat mass, no change in food intake, and a decrease in total energy expenditure (*Figure 2E–H*). This

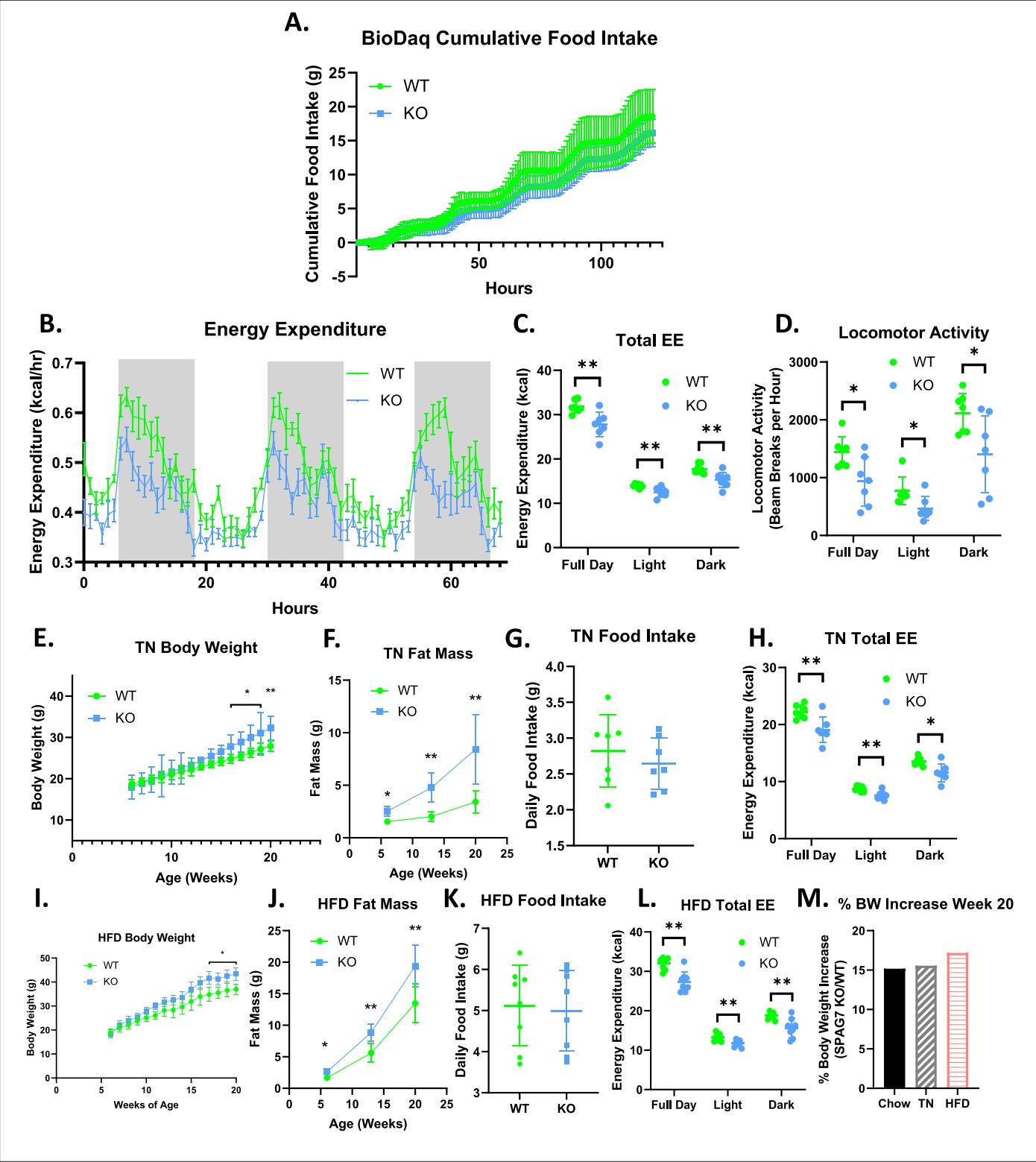

**Figure 2.** SPAG7-deficiency causes decreased locomotor activity and total energy expenditure. (**A**) Cumulative food intake as determined by BioDaq Food and Water intake monitoring system. N=7. (**B**) Hourly energy expenditure as determined by CLAMS metabolic cage system. N=7. (**C**) Total energy expenditure as determined by CLAMS metabolic cage system. n=7. Significance was assessed by Welch's two sample t-test. (**D**) Home cage locomotor activity as determined by CLAMS metabolic cage system. n=7. Significance was assessed by Welch's two sample t-test. (**E**) Body weight over time of WT vs SPAG7 KO animals raised at thermoneutrality. n=7. Significance was assessed by Welch's two sample t-test. (**F**) Fat mass over time of animals at thermoneutrality. n=7. Significance was assessed by Welch's two sample t-test. (**G**) Daily food intake of animals at thermoneutrality. N=7. (**H**) Total energy

*Figure 2 continued on next page*

*Figure 2 continued*

expenditure of animals at thermoneutrality as determined by CLAMS. n=7. Significance was assessed by Welch's two sample t-test. (**I**) Body weight over time of WT vs SPAG7 KO animals raised on high-fat diet. n=8. Significance was assessed by Welch's two sample t-test.(**J**) Fat mass over time of animals fed high-fat diet. n=8. Significance was assessed by Welch's two sample t-test. (**K**) Daily food intake of animals fed high-fat diet. N=8. (**L**) Total energy expenditure of animals fed high-fat diet. n=8. Significance was assessed by Welch's two sample t-test. (**M**) Percent body weight difference in SPAG7 KO animals vs WT fed chow diet at room temperature (Chow), fed chow diet at thermoneutrality (TN), or HFD at room temperature (HFD) at 20 weeks of age. n=7. * p<0.05, ** p<0.01.

indicates that thermogenesis is unlikely to play a significant role in the decreased energy expenditure and weight gain observed in the animals. SPAG7-deficient animals fed a 60% high-fat diet (HFD) also display the same phenotypes with no increase in food intake in SPAG7 KOs compared to WTs (*Figure 2I–L*). The percent body weight gain in SPAG7-deficient animals vs WT was very similar at room temperature on normal chow, at thermoneutrality on normal chow, and at room temperature on HFD: an increase of 15–17% (*Figure 2M*), indicating that decreased energy expenditure due to reduced locomotor activity is the key driver for the obesity phenotype in the SPAG7 KO mice.

## SPAG7-deficient animals display decreased exercise capacity and muscle function

The observation of reduced locomotor activity in the SPAG7 KO mice prompted us to investigate if there are defects in exercise performance and muscle function in these mice. WT and SPAG7 KO animals were subjected to treadmill endurance tests; the KO mice ran a significantly shorter distance than the WT animals before becoming exhausted (*Figure 3A*). Additionally, the maximum oxygen consumption reached during the treadmill endurance test was significantly reduced in SPAG7-deficient animals (*Figure 3B*). The maximum force produced by gastrocnemius complex hindlimb muscles when stimulated in anaesthetized animals was measured as indicative of muscle function and is severely impaired in SPAG7-deficient animals (*Figure 3C*). Skeletal muscle histology indicates a significant reduction in the cross-sectional area of Type IIa (fast-twitch) muscle fiber (*Figure 3D*, *Figure 3—figure supplement 1A*), indicating muscle atrophy. SPAG7-deficient skeletal muscle was also found to contain elevated levels of triglyceride content (*Figure 3E*). Skeletal muscle vascularization appeared normal (*Figure 3F–H*). We did, however, observe a marked reduction in oxidative capacity in skeletal muscle, as indicated by succinate dehydrogenase b activity staining and citrate synthase activity in gastrocnemius muscle (*Figure 3I and J*, *Figure 3—figure supplement 1B–C*).

Transcriptomics analysis of gastrocnemius muscle revealed 1,848 genes were significantly upregulated and 2,291 genes were downregulated in SPAG7 KO muscle compared to WT (*Figure 3K*, *Figure 3—figure supplement 2A–B*). Gene ontology enrichment analysis revealed many gene expression pathways that were affected by SPAG7-deficiency; among the most significantly downregulated were pathways involved in mitochondrial function (*Figure 3L*). The most significantly downregulated gene was *Hsd17b10*. This is a ubiquitously expressed mitochondrial protein that acts as both a short-chain dehydrogenase reductase and as a part of the RNase P complex, which is responsible for the maturation of all mitochondrial tRNA (*Holzmann et al., 2008*; *Vilardo et al., 2012*; *Yang et al., 2005*). Missense mutations in *Hsd17b10* occur in humans and result in HSD10 disease (*Chatfield et al., 2015*). HSD10 disease varies significantly in severity, but the most severe forms are characterized by developmental delays, decreased muscle tone, and neonatal mortality (*Ofman et al., 2003*; *Vilardo and Rossmanith, 2015*; *Zschocke, 2012*). *Hsd17b10* expression is reduced in nearly all SPAG7-deficient tissues (*Figure 3—figure supplement 2C–H*). Overall, these studies imply significant impairment of mitochondrial function and muscle performance in SPAG7-deficient animals.

## Induction of SPAG7-deficiency after early development does not cause obesity or skeletal muscle abnormalities

SPAG7 KO mice were smaller than WT mice during the neonatal stage and profoundly obese in adulthood (*Figure 1C*). This raised the question whether SPAG7-deficiency affects embryonic development, which may potentially cause the metabolic perturbations later in life. To test this hypothesis, an inducible whole-body SPAG7 KO (iSPAG7) mouse was developed. Using a globally expressed Cre-ERT2 along with a globally floxed *Spag7* gene, tamoxifen injection would induce global recombination and deletion of the *Spag7* gene (*Figure 4A*). To test the effects of SPAG7-deficiency in adulthood, iSPAG7

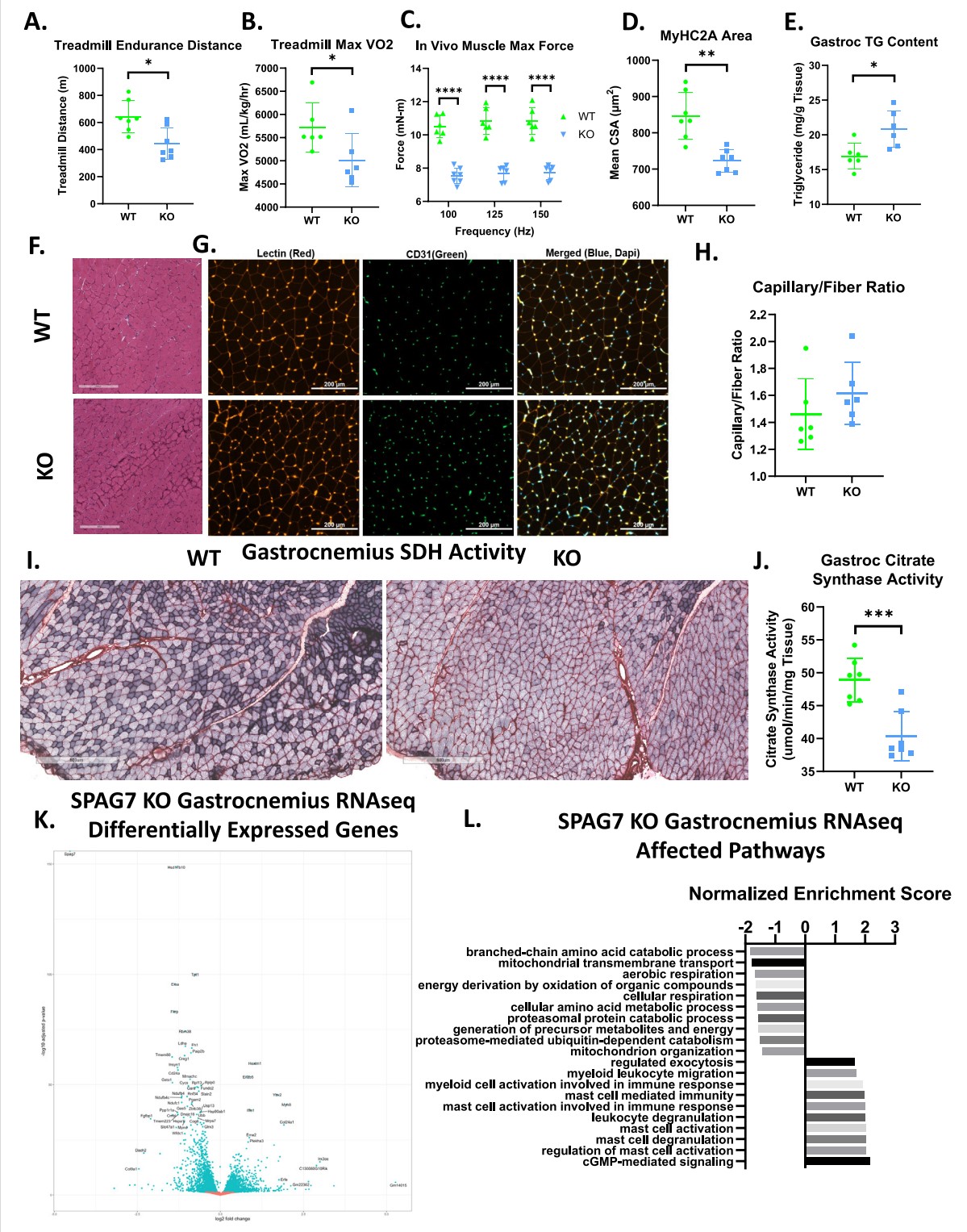

**Figure 3.** SPAG7-deficiency dampens skeletal muscle function and mitochondrial oxidative capacity. (**A**) Distance run until exhaustion for WT vs SPAG7 KO animals in treadmill endurance test. n=7. Significance was assessed by Welch's two sample t-test. (**B**) Max VO2 reached during treadmill endurance. n=6. Significance was assessed by Welch's two sample t-test. (**C**) In vivo gastrocnemius/soleus complex muscle max force generation. n=6. Significance was assessed by Welch's two sample t-test. (**D**) Cross sectional area of myosin heavy chain 2a-expressing fiber in gastrocnemius muscle. n=7. Significance was assessed by Welch's two sample t-test. (**E**) Triglyceride content of gastrocnemius muscle. n=6. Significance was assessed by Welch's two sample t-test. (**F**) Histological sections of gastrocnemius muscle stained with hematoxylin and eosin. Scale bars = 300 µm. (**G**) Histological

*Figure 3 continued on next page*

*Figure 3 continued*

sections of gastrocnemius muscle labeled with antibodies against lectin (red), CD31 (green), and DAPI (blue). Scale bars = 200 μm. (**H**) Quantification of CD31 +and lectin +capillaries per muscle fiber in gastrocnemius muscle. N=6. (**I**) Histological sections of gastrocnemius muscle stained for succinate dehydrogenase B activity. Scale bars = 600 μm. (**J**) Citrate synthase activity of gastrocnemius muscle. n=7. Significance was assessed by Welch's two sample t-test. (**K**) Volcano plot of differentially expressed genes in female WT vs SPAG7 KO gastrocnemius muscle following RNAseq. N=7. (**L**) Gene ontology enrichment pathway analysis of differentially expressed genes in female WT vs SPAG7 KO gastrocnemius muscle following RNAseq. n=7. * p<0.05, ** p<0.01, *** p<0.001, **** p<0.0001.

The online version of this article includes the following figure supplement(s) for figure 3:

**Figure supplement 1.** SPAG7-deficient muscle fiber staining.

**Figure supplement 2.** SPAG7-deficient skeletal muscle transcriptomics.

animals were aged to 7 weeks before being dosed with tamoxifen. In this manner, tamoxifen dosing was able to establish effective knockout of SPAG7 protein levels in all tissues examined, including liver, brain, kidney, muscle, and fat (*Figure 4B*). In contrast to the germline SPAG7 KO, iSPAG7 KO animals displayed no differences in body weight, lean mass, fat mass, or food intake (*Figure 4C–F*). There is no difference in glucose tolerance and total energy expenditure between the iSPAG7 KO and the WT mice (*Figure 4G–J*). Home cage locomotor activity, hindlimb max force generation, and treadmill endurance were all normal, as well (*Figure 4K–M*). These results indicate that the metabolic phenotypes are downstream of the effects of SPAG7-deficiency on the developing animal, suggesting that SPAG7 may play a role during embryonic development.

## SPAG7-deficiency causes placental insufficiency and intrauterine growth restriction

SPAG7-deficient mice are born significantly underweight (*Figure 5A*. 5B). They are also born below the expected rate. In heterozygous x heterozygous breeding, Mendelian ratios would predict roughly 25% of pups born to be SPAG7-deficient. We observe roughly 10% of pups to be SPAG7-deficient, indicating significant loss of SPAG7-deficient animals prior to birth (*Figure 5C*). Throughout mouse (and human) development, there are key stages where significant loss of embryos is more likely to occur. These include fertilization (e0), gastrulation (e6.5-e8), organogenesis (e8-e11.5), and birth (e19-e20). Genotyping embryos before and after each of these developmental stages, we observe expected Mendelian ratios of SPAG7-deficient animals from e6.5 through e11.5, indicating no loss of embryos during fertilization, gastrulation, and organogenesis (*Figure 5D–F*). We do see a significant decrease in SPAG7-deficient percentage at e18.5, indicating that SPAG7-deficient embryos are lost at some point after e11.5 and prior to birth (*Figure 5G*). However, it should be noted that SPAG7-deficient embryos appear to be smaller and less well-developed than WT embryos within the same dam, at each of these stages.

At e18.5, SPAG7-deficient animals display decreased fetal and placental weight (*Figure 5H–J*), indicating intrauterine growth restriction (IUGR). Other than the smaller size, the SPAG7-deficient fetus lacks any specific morphologic abnormalities: all organs and limbs appear appropriately developed for their size. To further characterize the IUGR phenotype, we measured markers relevant to IUGR in the fetus. SPAG7-deficient fetuses have lower blood glucose and plasma insulin compared with WT controls (*Figure 5K and L*). Insulin-like growth factor (IGF) signaling has been implicated as essential for fetal growth and is dysregulated in IUGR fetuses (*Agrogiannis et al., 2014*; *Reid et al., 2002*). SPAG7-deficient fetal liver displays decreased IGF1 and 2 protein and gene expression levels, indicating reduced IGF signaling (*Figure 5M–O*). These findings are consistent with an intrauterine growth restriction phenotype (IUGR) caused by placental insufficiency and suggest SPAG7 may be necessary to maintain placental function.

SPAG7 is expressed highly in mouse and human placenta (*Uhlén et al., 2015*), it is also highly expressed in every mouse tissue during the earliest stages of development (*Baldarelli et al., 2021*; *Brunskill et al., 2014*; *Schmitt et al., 2014*). SPAG7-deficient placenta is smaller in cross-sectional area compared to the placenta of WT littermates (*Figure 6A–C*). SPAG7-deficient placenta displayed no significant changes in vascularization as measured by CD34 IHC staining and gene expression of vascularization markers (*Figure 6B, D and E*). There is a significant increase in mitochondrial DNA content in SPAG7 KO placenta (*Figure 6F*). This finding is consistent with previous findings in IUGR

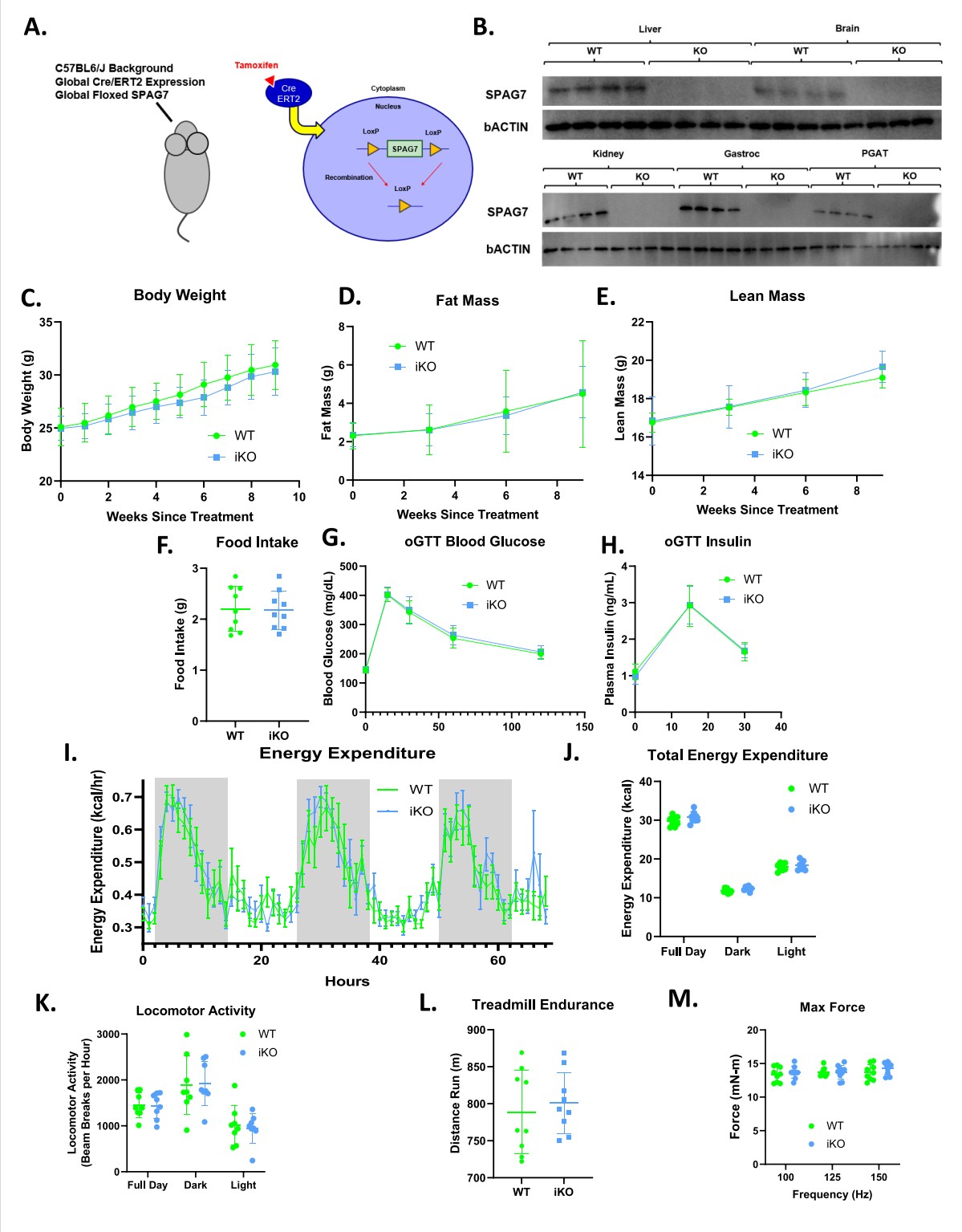

**Figure 4.** Whole-body SPAG7-deficiency induced during adulthood has no effect on systemic metabolism. (**A**) Graphic representation of the whole-body inducible SPAG7-deficient mouse model (iSPAG7 KO). (**B**) Western blot for SPAG7 and bACTIN in liver, brain, kidney, gastrocnemius muscle, and PGAT tissues from iSPAG7 KO animals, 8 weeks following final tamoxifen dose. N=4. (**C**) iSPAG7 KO body weight over time. N=9. (**D**) iSPAG7 KO fat mass over time. N=9. (**E**) iSPAG7 KO lean mass over time. N=9. (**F**) iSPAG7 KO daily food intake, as measured by hopper weight. Taken 9 weeks after treatment. N=9. (**G**) Blood glucose levels following an oral glucose bolus. Taken 7 weeks after treatment. N=9. (**H**) Plasma insulin levels following an oral glucose bolus. Taken 7 weeks after treatment. N=9. (**I**) Hourly energy expenditure as determined by CLAMS metabolic cage system. Taken 6 weeks

*Figure 4 continued on next page*

*Figure 4 continued*

after treatment. N=9. (**J**) Total energy expenditure as determined by CLAMS metabolic cage system. Taken 6 weeks after treatment. N=9. (**K**) Home cage locomotor activity as determined by CLAMS metabolic cage system. Taken 6 weeks after treatment. N=9. (**L**) Distance run until exhaustion during treadmill endurance test. Taken 8 weeks after treatment. N=9. (**M**) In vivo gastrocnemius/soleus complex muscle max force generation. Taken 8 weeks after treatment. n=9.

The online version of this article includes the following source data for figure 4:

**Source data 1.** Uncropped western blot gels.

and is generally thought to be a compensatory response (*Mandò et al., 2014*; *Naha et al., 2020*). Histological analyses showed that SPAG7 KO placenta displayed abnormal morphology, exhibiting disruption of both the labyrinth and junctional zones. The junctional zone, however, was particularly severely affected, as demonstrated by decreased junctional:labyrinth zone ratios and percent junctional zone (*Figure 6G–I*). The junctional zone of the placenta plays a key role in the production of hormones and the transport of metabolites from dam to pup (*Woods et al., 2018*). A disruption in this area is likely to reduce nutrient availability to the fetus.

## Discussion

We identified mutations in the gene *spag7* that induce obesity and insulin resistance in mice from a mouse forward genetic screen. The biological function of SPAG7 is largely unknown. Utilizing constitutive and inducible SPAG7 knockout mouse models, we investigated the function of SPAG7 in vivo. The results show that SPAG7-deficient mice were born with lower body weight; however, they developed obesity and insulin resistance in adulthood. The metabolic disturbance in the SPAG7 KO animals is due to a decrease in energy expenditure, driven by decreased locomotor activity. Skeletal muscle function and exercise endurance in SPAG7-deficient animals are significantly impaired, displaying decreased force generation and oxidative capacity. In addition, these phenotypes are downstream of developmental effects, as knocking out SPAG7 in adulthood did not cause these metabolic disturbances. Furthermore, SPAG7 KO embryos display intrauterine growth restriction (IUGR), and aberrant development of the junctional zone of SPAG7-deficient placenta. These results demonstrate that SPAG7 is essential for fetal growth and placental function. Lack of SPAG7 results in IUGR, which leads to the development of obesity, insulin resistance, and other metabolic disturbances later in life.

Fetal nutritional status is well-known to drive metabolic health in adulthood. Perhaps the most famous example of this is the Dutch Hunger Winter. Pregnant women, exposed to acute and extreme decreases in nutritional intake, gave birth to low birthweight newborns, who were then at higher risk for developing obesity and diabetes in adulthood (*Roseboom et al., 2006*). Intrauterine growth restriction (IUGR) has been linked to the development of metabolic syndrome later in life. However, the mechanisms that cause IUGR and influence its association with adult metabolic health remain poorly understood. Bilateral uterine artery ligation during gestation in rats leads to decreased birth weights, but obesity in adulthood (*Simmons et al., 2001*). Placental insufficiency is a common cause of IUGR (*Malhotra et al., 2019*; *Wood, 2020*). In fact, junctional zone deficiency, in particular, has been shown to negatively affect fetal growth and birth weight (*Mark et al., 2011*; *Woods et al., 2018*).

IUGR has been shown to affect skeletal muscle health and performance. Humans that experience IUGR, for any given population, display decreased skeletal muscle mass that is not compensated for after birth and persists through adulthood (*Năstase et al., 2018*). The skeletal muscle of IUGR fetuses in sheep display decreased intracellular ATP content and decreased amino acid uptake (*Sarkar et al., 2014*; *Stremming et al., 2020*). Muscle mitochondrial oxidative capacity can also be lower in IUGR skeletal muscle (*Stremming et al., 2022*).

The deleterious effects of SPAG7-deficiency on the fetus clearly indicate an important function for SPAG7 during embryonic development. The most common cause for IUGR is placental insufficiency. In the rodent, the placenta can be roughly broken up into three zones: the decidua, the labyrinth zone, and the junctional zone. The decidua and labyrinth zones contain the vasculature of the mother and fetus, respectively. The junctional zone is the main site of nutrient and gas exchange and also the main endocrine compartment of the placenta, producing hormones, growth factors, and cytokines that are important for the normal progression of pregnancy and fetal growth (*Woods et al., 2018*). Histological examination of SPAG7 KO placenta revealed abnormal morphology of both the labyrinth

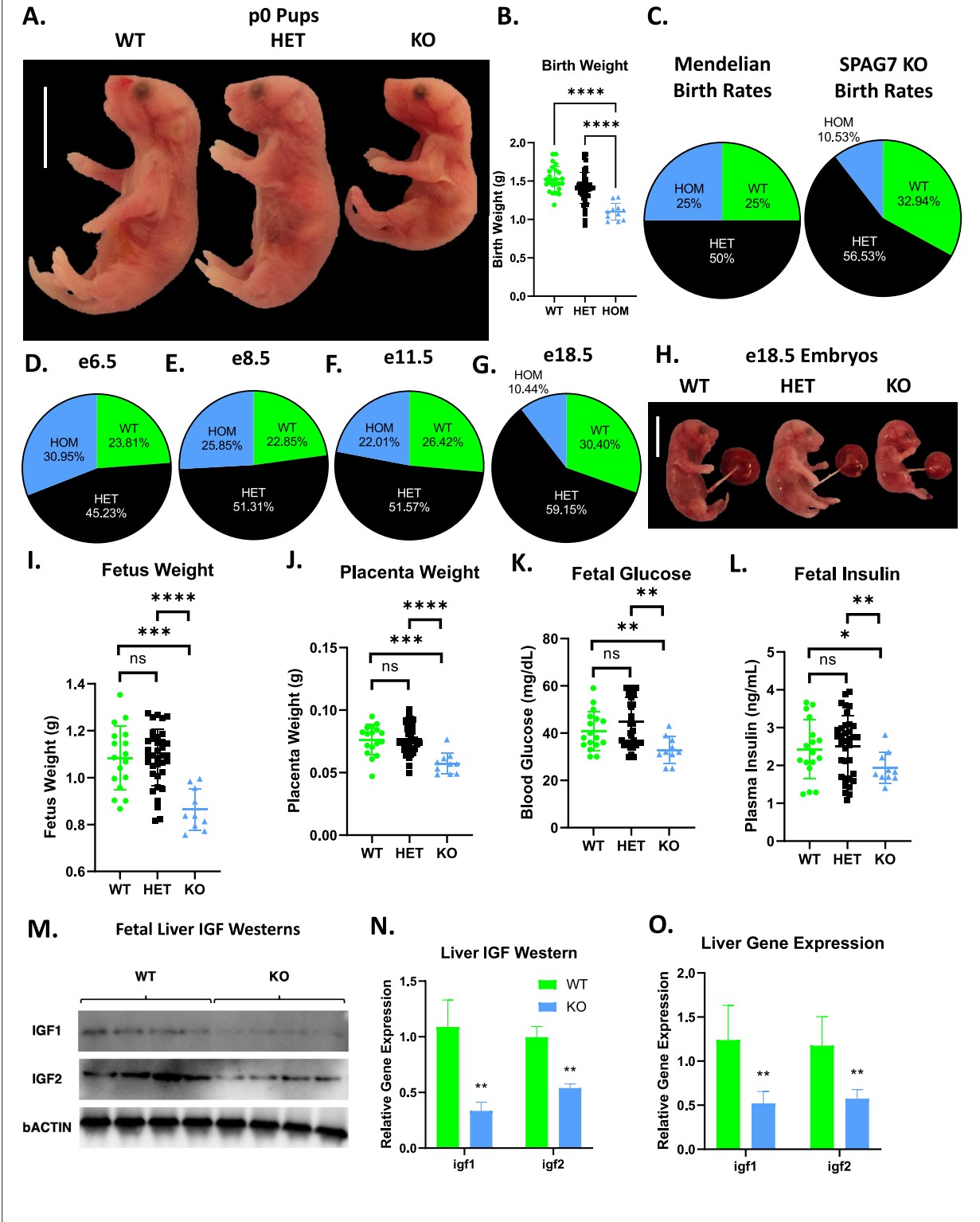

**Figure 5.** SPAG7-deficiency induces intrauterine growth restriction. (**A**) Gross morphology of WT, SPAG7 KO Heterozygous, and SPAG7 KO Homozygous pups at p0. Scale bars represent 10 mm. (**B**) Birth weights of WT, SPAG7 KO Heterozygous, and SPAG7 KO Homozygous pups. Number of WT pups = 25. Number of HET pups = 45. Number of HOM pups = 11. Significance was assessed by one-way ANOVA and Tukey HSD. (**C**) Mendelian birth genotyping rates expected from Heterozygous x Heterozygous breeding (left), and birth genotyping rates observed from SPAG7 KO Heterozygous x Heterozygous breeding (right). Number of dams = 29. (**D**) Embryo genotyping rates at e6.5 from SPAG7 KO Heterozygous x Heterozygous breeding. Number of dams = 6. (**E**) Embryo genotyping rates at e8.5 from SPAG7 KO Heterozygous x Heterozygous breeding. Number of dams = 6. (**F**) Embryo

*Figure 5 continued on next page*

Figure 5 continued

genotyping rates at e11.5 from SPAG7 KO Heterozygous x Heterozygous breeding. Number of dams = 6. (**G**) Embryo genotyping rates at e18.5 from SPAG7 KO Heterozygous x Heterozygous breeding. Number of dams = 6. (**H**) Gross morphology of WT, HET, and SPAG7 KO embryos at e18.5. Scale bars represent 10 mm. (**I**) Fetus weights at e18.5, comparing WT and SPAG7 KO embryos. n=10. Significance was assessed by one-way ANOVA and Tukey HSD. (**J**) Placenta weights at e18.5, comparing WT and SPAG7 KO embryos. n=10. Significance was assessed by one-way ANOVA and Tukey HSD. (**K**) Fetal blood glucose levels at e18.5, comparing WT and SPAG7 KO embryos. n=10. Significance was assessed by one-way ANOVA and Tukey HSD. (**L**) Fetal plasma insulin levels at e18.5, comparing WT and SPAG7 KO embryos. n=10. Significance was assessed by one-way ANOVA and Tukey HSD. (**M**) Western blots for bACTIN, IGF1, and IGF2 in WT and SPAG7 fetal liver at e18.5. n=4. (**N**) Quantification of western blots in *Figure 6M*. n=4. Significance was assessed by Welch's two sample t-test. (**O**) Gene expression levels of *igf1* and *igf2* in fetal liver at e18.5. n=6. Significance was assessed by Welch's two sample t-test. * p<0.05, ** p<0.01, *** p<0.001, **** p<0.0001.

The online version of this article includes the following source data for figure 5:

**Source data 1.** Uncropped western blot gels.

and junctional zones. In addition, the junctional zone was markedly diminished in SPAG7 KO placenta, as demonstrated by decreased junctional:labyrinth zone ratios and decreased percent junctional zone (*Figure 6G–I*). The mature mouse placenta is established mid-gestation (~e10.5) and continues to grow, in size and complexity, throughout the pregnancy. Early in gestation, the yolk sac supports the nutritional needs of the embryo, however, from mid-gestation onward, the fetal nutrient supply requires the transport capacity of the placenta. The success of this transition is critical for fetal survival and continued growth. During our studies, we did not observe significant loss of SPAG7 KO embryos until after the switch to placental nutrition supply, which could be due to the disruption of labyrinth and junctional zone development in the placenta.

We hypothesize the SPAG7 KO fetuses that survive do so by decreasing the mitochondrial metabolic capacity of their skeletal muscle and other tissues. Reducing muscle energy consumption in response to the decreased nutrient availability in the IUGR fetus is advantageous to ensure fetal survival. However, when these animals are born and age, this reduction predisposes them towards metabolic disease (*Pendleton et al., 2021*). After birth, when nutrition is readily available, the decrease in energy expenditure, combined with food intake normalized to wild-type littermates, eventually leads to increased body weight and fat mass. It is worth noting that SPAG7 KO mice are not hyperphagic, a phenotype observed in some IUGR models, suggesting that other perturbations in SPAG7 KO animals may counter-regulate hyperphagia. The key driver for the development of obesity and metabolic syndrome in the SPAG7 KO mouse is reduced energy expenditure. Models that produce obesity, primarily driven by decreased locomotor activity, are rare. Further investigation of SPAG7-deficient mice may provide valuable information for drugs targeting energy expenditure in the treatment of obesity.

Overall, our results demonstrate that SPAG7 plays an important role in fetal development. SPAG7 is essential for the proper formation of the junctional and labyrinth zones of the placenta, which contributes to IUGR in the SPAG7 KO fetus (*Figure 7*). The surviving SPAG7 KO mice develop a range of metabolic disorders including obesity, insulin resistance, glucose intolerance, exercise impairment, reduced energy expenditure, and reduced skeletal muscle mitochondrial oxidative capacity in adulthood. Our results uncover, for the first time, the biological function for the *Spag7* gene in vivo. Furthermore, these findings shed light into a new biological pathway regulating fetal development. The results provide insights into how skeletal muscle function contributes to systemic energy balance and how fetal nutritional status can have far-reaching effects, influencing the nutritional status of the adult. Deeper understanding of the mechanisms involved in placental insufficiency, IUGR, and SPAG7 biology may give us the opportunity to reverse aberrations in fetal growth rate and prevent the metabolic co-morbidities associated with low birth weight (*Wood, 2020*).

## Materials and methods
### Mice

All animal experiments were conducted following study protocols and procedures reviewed and approved by Pfizer Institutional Animal Care and Use Committee. The facilities that supported this work are fully accredited by AAALAC International.

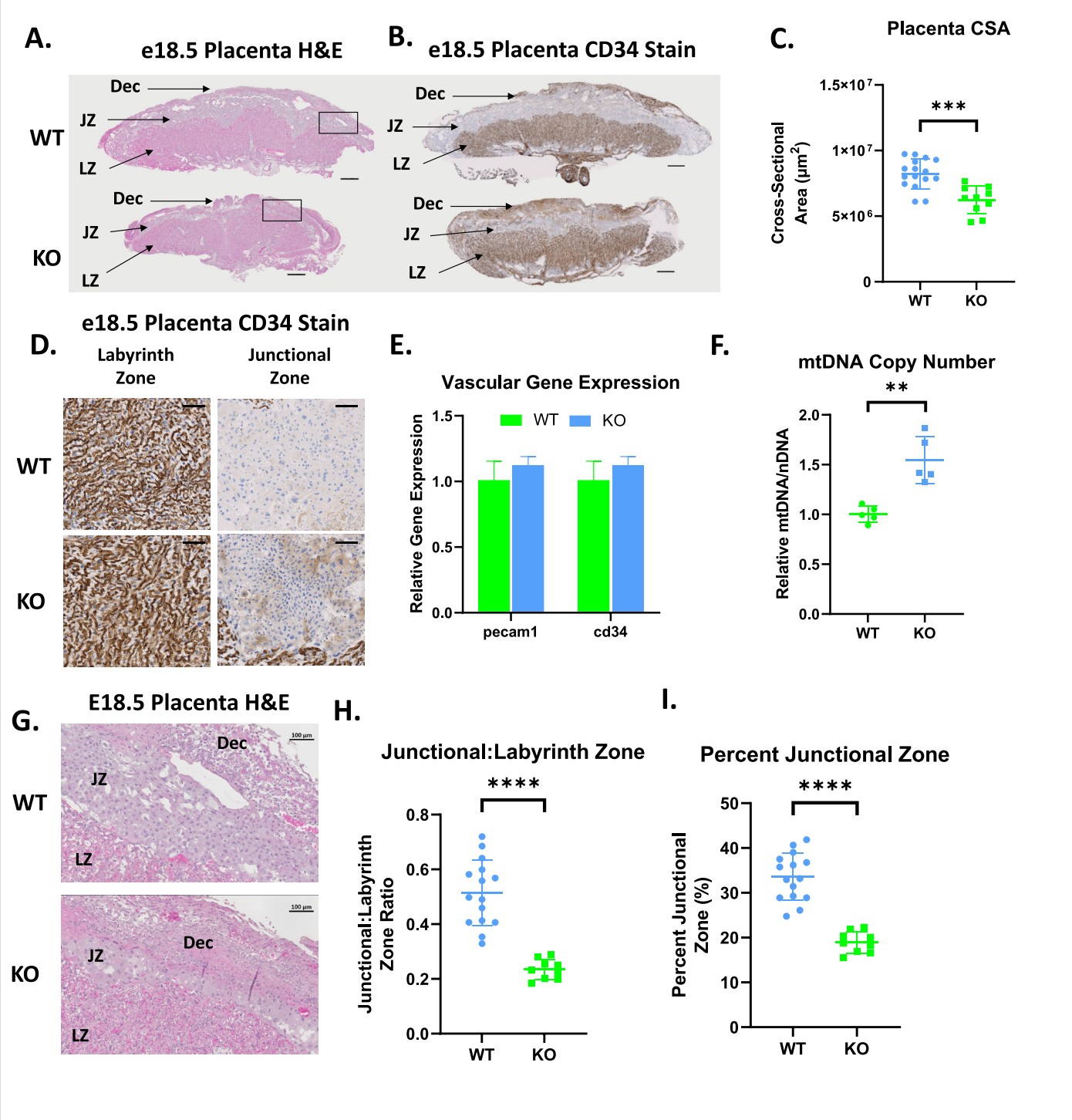

**Figure 6.** SPAG7-deficiency induces placental insufficiency. (**A**) Histological sections of WT and SPAG7 KO placenta stained with hematoxylin and eosin. Scale bars = 500 um. (**B**) Histological sections of WT and SPAG7 KO placenta labeled with CD34 antibody. Scale bars = 500 um. (**C**) Total placenta cross-sectional area of WT and SPAG7 KO placenta at e18.5. n=9. Significance was assessed by Welch's two sample t-test. (**D**) Histological sections of WT and SPAG7 KO placenta labyrinth and junctional zones labeled with CD34 antibody. Scale bars = 100 um. (**E**) Whole placenta gene expression of vascular markers. N=5. (**F**) Whole placenta mtDNA copy number. n=5. Significance was assessed by Welch's two sample t-test. (**G**) Histological sections of WT and SPAG7 KO placenta stained with hematoxylin and eosin. Scale bars = 100 um. (**H**) Junctional:Labyrinth Zone ratios in WT and SPAG7 KO placenta. n=9. Significance was assessed by Welch's two sample t-test. (**I**) Percent Junctional Zone in WT and SPAG7 KO placenta. n=9. Significance was assessed by Welch's two sample t-test. * p<0.05, ** p<0.01, *** p<0.001, **** p<0.0001.

# Graphical Abstract

**A.**

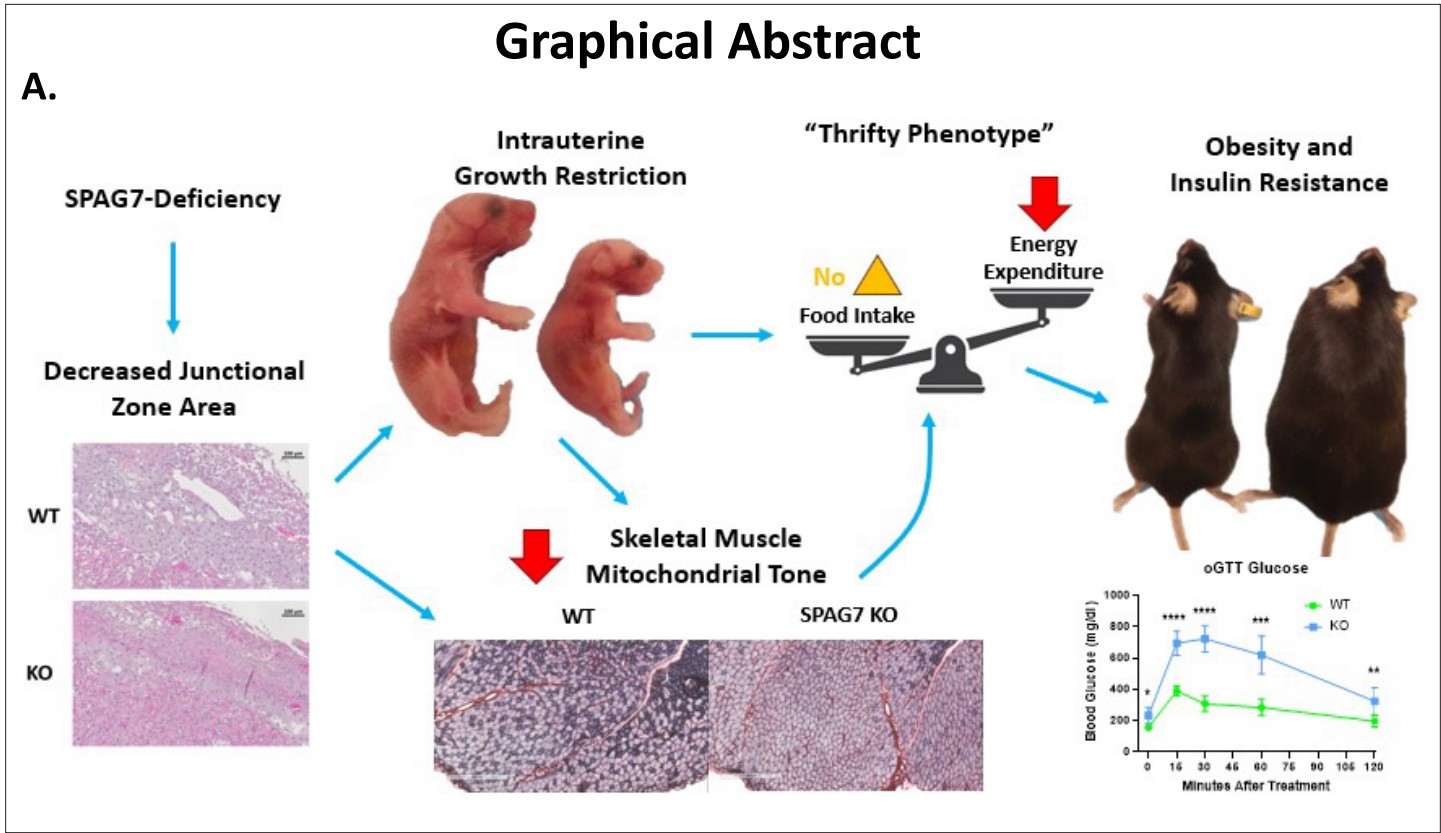

**Figure 7.** Graphical abstract. (**A**) SPAG7-deficiency causes intrauterine growth restriction which leads to obesity and insulin resistance in adulthood.

Adult female wild-type (WT), SPAG7 KO, (C57BL/6 J; various ages) were obtained from Jackson Laboratories (Farmington, CT). Mice were housed individually in Innovive cages (Innorack IVC Mouse 3.5) under a standard 12 hr light/12 hr dark cycle (06:00 hr: 18:00 hr) in a temperature- and humidity-controlled environment (22 ± 1°C for room temperature, 28 ± 1°C for thermoneutrality). Mice were given ad libitum access to tap water and standard chow (Innovive mouse metal feeder; Purina rodent diet 5061; Purina Mills, St. Louis, MO) or 60% high-fat diet (Research Diets D12492, New Brunswick, NJ). Body weight was recorded weekly (Mettler Toledo ME4002TE, Mettler Toledo, Oakland, CA). Body composition was assessed using the EchoMRI 4-in-1 500 Body Composition Analyzer (EchoMRI, Houston, TX).

## Generation of *Spag7* cKO mouse model using the CRISPR/Cas9 technology

The *Spag7* cKO model was created on the C57BL/6 J strain (JAX #000664) by inserting two LoxP sites flanking the exon 2 of the gene in the genome utilizing the CRISPR/Cas9 technology (*Wang et al., 2013*). One LoxP is positioned at 316 bp upstream of the exon 2 and the other at 97 bp downstream of the exon 2. A donor plasmid used as a repair template consists of the floxed genomic sequence flanked by 0.7 kb 5' homology arm and 0.4 kb 3' homology arm. Protospacer sequences of the two sgRNAs targeting the upstream and downstream insertion sites are 5'-cctgaagccagtgtat attg and 5'-gggctttagttccaccatc, respectively. These sgRNAs were synthesized by in vitro transcription. For genome editing in mouse zygotes, Cas9 protein (IDT, #1081058), sgRNAs and the donor plasmid were mixed and microinjected into 1 cell embryos harvested from super-ovulated donors. Injected embryos were transferred to pseudo-pregnant females to generate live pups. For genotyping of founders and their offspring, genomic PCR was performed with two primers external to the 5' and 3' homology arms (5'- TCACATCACGGTCCATCATC and 5'- TCATGACATAGCGACAGTCA, respectively) and amplified products were sequenced to confirm correct editing. Selected founders were bred to wildtype C57BL/6 J mice for germline transmission of the cKO allele.

## Western blotting

Animal tissues were snap-frozen in liquid nitrogen and stored at −80 °C. Frozen tissues were homogenized in ice-cold RIPA buffer (Sigma Aldrich, St. Louis, MO), and protein concentrations were determined using a BCA protein assay (Thermo Fisher Scientific, Waltham, MA). Protein extracts were separated on NuPAGE 4–20% Bis-Tris gels (Bio-Rad, Hercules, CA) and blotted onto PVDF membranes (Bio-Rad, Hercules, CA). Membranes were blocked for 1 hr at room temperature in TBST (0.1%) containing 5% milk. Membranes were then incubated overnight at 4 °C with primary antibodies. Following three washing steps with TBST (0.1%), membranes were incubated with HRP-conjugated secondary antibodies for 1 hr at room temperature. After thorough washing, proteins were visualized with SuperSignal West Dura Extended Duration Substrate (Thermo Fisher Scientific, Waltham, MA) on the Bio-Rad ChemiDoc MP. Immunoreactive bands were quantified using Image J Software (NIH). Primary antibodies used: Anti-SPAG7 (Sigma, HPA024032), Anti-IGF1 (ThermoFisher, MA5-35060), Anti-IGF2 (ThermoFisher, PA5-71494), and Anti-bACTIN (ThermoFisher, MA5-32540).

## Quantitative RT-PCR

For mouse studies, animals were euthanized via $CO_2$ and tissues were immediately dissected and snap frozen in liquid nitrogen. Ribonucleic acid (RNA) was extracted and purified using RNeasy RNA Isolation Kit (QIAGEN, Germantown, MD) then reverse transcribed into complementary deoxyribonucleic acid (cDNA) with a High-Capacity cDNA Reverse Transcription Kit (Applied Biosystems, Foster City, CA). Quantitative reverse transcriptase polymerase chain reaction (qRT-PCR) was performed using TaqMan reagents and primer-probes (Applied Biosystems, Foster City, CA). Gene expression was normalized to the control gene TATA box binding protein (*Tbp*).

## Bone mineral density and body length

Dual X-ray absorptiometry (DEXA) imaging was used to assess body composition, bone mineral density and body length of SPAG7 knock out and litter mate or age-matched control mice. The imaging was conducted on the Ultra-Focus system Faxitron (BioVision). The system was calibrated and warmed up prior to image acquisition. Animals were anesthetized using isoflurane anesthesia (1 to 4%), and imaged in the prone position, with paws stretched and taped to maintain position. Whole body images were captured using both energies (40 and 80 kVP). Body composition was measured from DEXA images of each animal using software embedded tools (Vison DEXA). Overall length of each was animal measured from tip of mouse nose to the distal end of the tail using in built DEXA image analysis tool. Screenshots of data were saved and later transcribed to an excel sheet.

## Circulating lipid and liver enzyme analysis

At 13 weeks of age, SPAG7 KO animals were bled via tail nick and blood was collected in EDTA-coated tubes (Becton, Dickinson and Company, Franklin Lakes, NJ) and centrifuged to collect plasma (2000 × *g* for 10 min at 4 °C), which was stored at −80 °C until analysis. For fasted measurements, animals were bled following a 16 hr overnight fast. For refed measurements, animals were bled following a 2 hr refeed following a 16 hr overnight fast. Plasma was analyzed using the ADVIA Chemistry Analyzer XPT/1800 (Siemens, Munich, Germany). Plasma triglycerides (ADVIA XPT, Siemens, Munch Germany), cholesterol (ADVIA XPT, Siemens, Munch Germany), and non-esterified fatty acids (ADVIA XPT, Siemens, Munch Germany) were analyzed in all mice.

## In vivo metabolic studies

All metabolic tests were performed using standard protocols on female littermates (aged: 6–32 weeks). Mice were acclimated to handling and injection (0.1 mL saline) or oral gavage (0.1 mL saline) stress for ~5 days prior to the start of all studies. For glucose tolerance tests (GTTs), animals were fasted for 8 hr with ad libitum access to water. Fasting glucose measurements were taken before 2 mg/kg oral glucose dose. Blood glucose measurements were taken via tail nick using the AlphaTRAK 2 Glucose Monitoring System (Zoetis, Parsippany-Troy Hills, NJ) at 15, 30, 60, and 120 min after dose. Blood was collected for insulin measurement in EDTA tubes (Becton, Dickinson and Company, Franklin Lakes, NJ) and centrifuged to collect plasma (2000 × *g* for 10 min at 4 °C), which was stored at −80 °C until analysis at 0, 15, and 30 min after dose. Insulin levels were measured using an Ultrasensitive Mouse Insulin ELISA (ALPCO, Salem, NH), according to manufacturer's instructions. For insulin tolerance tests

(ITTs), animals were fasted for 8 hr and blood glucose was measured before (0 min) and at 15, 30, 60, and 120 min following IP injection of.75 IU/kg of insulin (Humulin R, Eli Lilly, Indianapolis, IN).

## Adipose tissue analysis

At 32 weeks of age, SPAG7 KO animals were sacrificed, perigonadal/visceral, inguinal/subcutaneous, and intrascapular/brown adipose tissues were removed and weighed. They were then fixed in 10% formalin, processed to paraffin, sectioned at 5 µm, stained with H&E, and scanned at ×40 magnification using the AxioScan7 (Zeiss, Jena, Germany). Two 1000 µm x 1000 µm samples from homogeneous adipocyte areas of each image were sent to ImageJ and further analyzed with the Adipocyte Tools plugin using the option for large cells and the settings of minimum size 500 µm$^2$ and number of erosions rounds 3. Faulty detections were manually corrected. Cell diameter was calculated assuming circular shape of adipocytes. Crownlike structures (CLS) were identified in subcutaneous AT sections as single adipocytes surrounded by at least 4 F4/80-positive macrophages.

## Triglyceride content assays

32-week-old WT and SPAG7 KO animals were taken down and gastrocnemius muscle and liver were dissected and snap-frozen in liquid nitrogen. 50–100 mg of pulverized tissue was homogenized and analyzed for triglyceride content using the Roche Hitachi 912 Analyzer Triglyceride Reagent Set (GMI, Ramsey, MN).

## Food intake

Food intake measurements were made using in-cage hopper weights or in-cage ceramic bowls, measured weekly. Food intake was further monitored using the BioDAQ food and water intake monitoring system system (Research Diets, New Brunswick, NJ). Mice were acclimated for four days in the BioDAQ system before food intake experiments were run. Food intake was monitored continuously and recorded at time intervals indicated in the figures.

## Metabolic cage studies

Energy expenditure (EE), and locomotor activity were measured with an Oxymax indirect calorimetry system and CLAMS monitoring system (Columbus Instruments, Columbus, OH) starting at 11 weeks of age. The animals were given free access to water and diet. Means were calculated for dark, light, and full day cycles. Animals were acclimatized to the cages for 48 hr prior to data collection.

## Treadmill endurance test

Mice were progressively acclimatized to treadmill exposure by increasing intensity and duration for 3 days. For the incremental test, mice ran on an enclosed, single lane treadmill (molecular enclosed metabolic treadmill for mice; Columbus Instruments), and real-time measurements of oxygen consumption ($VO_2$) and carbon dioxide output ($VCO_2$) were performed using an Oxymax/Comprehensive Laboratory Animal Monitoring System (CLAMS; Columbus Instruments). Mice ran at 6, 9, 12, 18, 21, and 23 m·min$^{-1}$ for 3 min at each velocity at a 0°, 5°, 10°, and 15° inclination for 6 min at each inclination. Exhaustion was defined as the inability to return to treadmill running after 10 s.

## In vivo muscle max force generation

Mice were anaesthetized with isoflurane and placed supine on a platform heated via a circulating water bath at 37 °C. The right leg was shaved up to the patella and right knee stabilized via knee clamp. Once stabilized, the right foot was affixed to a Dual Mode Foot Plate (300 C FP, Aurora Scientific Inc, Aurora, Canada), and two electrodes were placed subcutaneously near the mid-belly of the gastrocnemius to achieve plantar flexion. A 1 Hz electrical stimulation was delivered (0.2 s duration, 1 s between stimulations) via stimulator (701 C, Aurora Scientific Inc) while increasing amperes to generate a maximum twitch measurement. After a maximum twitch was established, a force frequency of isometric contractions was initiated (0.2 s duration, 120 s between stimulations). All data were collected and analysed using the manufacturer supplied software (DMC and DMA, Aurora Scientific Inc). Mice were housed at thermoneutral temperature prior to analysis.

## Histology and immunohistochemistry

Formalin fixed adipose tissue, livers, and gastrocnemius muscles were dehydrated and embedded in paraffin. One paraffin-embedded cross-section (5 μm thickness) through the adipose tissue, the left lateral lobe of the liver, and the mid-body of the gastrocnemius muscle from each animal was stained with hematoxylin and eosin using an automatic slide stainer (Leica) and microscopic evaluation was performed by a veterinary pathologist.

For muscle vascularity staining, mouse gastrocnemius muscle was isolated, mounted on a cork and snap-frozen in cold isopentane (Sigma-Aldrich, 277258) for cryo-sectioning. Cross-sections of 10 μm thickness were collected from the belly of the samples and placed on Superfrost Plus microscope slides. For capillary density and fiber size evaluation, the sections were stained with CD31 antibody (Biocare, CM-303A) and Dylight 594 tomato lectin (Vector Laboratories, DL-1177) for 1 hr under room temperature followed by an 1 hr incubation of an alexa 488 goat anti-rat secondary antibody (Thermo Fisher, A-11006). The slides were mounted with VECTASHIELD Antifade Mounting Medium with 4',6-diamidino-2-phenylindole (DAPI) (Vector Laboratories, H-1800) and were imaged using a Zeiss AxioScan Z.1 slide scanner (Carl Zeiss, Jena, Germany). The images were analyzed using Visiopharm (Version 2020.09.0.8195) and custom-designed applications. The capillary/fiber ratio was calculated by dividing the total count of capillary (CD31+ lectin+) by the total count of fiber in the cross-section. The fiber size was the median surface area of all fibers in the cross-section.

For placenta vascularity staining, paraffin-embedded sections (5 μm thickness) were deparaffinized and rehydrated. Sections were blocked in 5% goat serum for 1 hr at room temperature. Sections were incubated with anti-CD34 primary antibody (Abcam ab81289) at a 1:200 concentration for 1 hr at room temperature and then overnight at 4 degrees. Sections were washed with PBS before incubation with secondary antibody for 1 hr at room temperature. Sections were washed again and imaged using a Zeiss AxioScan Z.1 slide scanner (Carl Zeiss, Jena, Germany).

## Skeletal muscle SDH activity stain

Transverse sections (5 μm) were cut from the mid-body of the gastrocnemius muscle. The sections were dried at room temperature for 30 min before incubation in a solution made up of 0.2 M phosphate buffer (pH 7.4), 0.1 M $MgCl_2$, 0.2 M succinic acid (Sigma Chemical Company, St. Louis, MO, USA) and 2.4 mM nitroblue tetrazolium (NBT, Sigma) at 37 °C in a humidity chamber for 45 min. The sections were then washed in deionized water for three minutes, dehydrated in 50% ethanol for two minutes, and mounted for viewing with DPX mount medium (Electron Microscopy Sciences, Hatfield, PA, USA). Digital photographs were taken from each section at ×10 magnification under a Nikon Eclipse TE 2000-U microscope (Nikon, Melville, NY, USA) with a Nikon digital camera (Digital Sight DS-Fi1), and fibers were quantified with imaging software (Image J, NIH).

## Skeletal muscle citrate synthase activity

Whole gastrocnemius muscle was removed from the animals and snap-frozen in liquid nitrogen. The tissue was homogenized in CelLytic MT Cell Lysis Reagent (Millipore-Sigma) with protease inhibitors (Sigma). Citrate Synthase activity was determined using a Citrate Synthase Assay Kit (Millipore-Sigma), in accordance with manufacturer's instructions.

## RNAseq sample preparation

28-week-old WT and SPAG7 KO animals were sacrificed and gastrocnemius muscle was snap-frozen in liquid nitrogen. Tissue was pulverized and total RNA was extracted using Qiagen Rneasy Plus Universal mini kit following manufacturer's instructions (QIAGEN, Hilden, Germany). The RNA samples were quantified using Qubit 2.0 Fluorometer (Thermo Fisher Scientific, Waltham, MA, USA) and RNA integrity was checked using TapeStation (Agilent Technologies, Palo Alto, CA, USA). RNA sequencing libraries were prepared using the NEBNext Ultra Directional RNA Library Prep Kit for Illumina following manufacturer's instructions (NEB, Ipswich, MA, USA). Briefly, mRNAs were first enriched with Oligo(dT) beads. Enriched mRNAs were fragmented for 15 min at 94 °C. First strand and second strand cDNAs were subsequently synthesized. cDNA fragments were end repaired and adenylated at 3'ends, and universal adapters were ligated to cDNA fragments, followed by index addition and library enrichment by limited-cycle PCR. The sequencing libraries were validated on the Agilent TapeStation (Agilent Technologies, Palo Alto, CA, USA), and quantified by using Qubit 2.0

Fluorometer (Invitrogen, Carlsbad, CA) as well as by quantitative PCR (KAPA Biosystems, Wilmington, MA, USA).

## RNA sequencing

The sequencing libraries were multiplexed and clustered onto a Illumina flowcell. After clustering, the flow cell was loaded onto the Illumina NovaSeq 6000 instrument according to manufacturer's instructions. The samples were sequenced using a 2x150 bp Paired End (PE) configuration and targeting 30 million reads/sample. Image analysis and base calling were conducted by the Illumina Control Software (HCS). Raw sequence data (.bcl files) generated from Illumina were converted into fastq files and de-multiplexed using Illumina bcl2fastq 2.20 software. One mis-match was allowed for index sequence identification.

## Transcriptomic analysis

The raw FASTQ files were processed using an internal pipeline, Rnaseq Experiment Dashboard, utilizing STAR (v2.7.3a) (*Dobin et al., 2013*) and Salmon (v1.5.2) (*Patro et al., 2017*) to map reads to the mouse genome and transcriptome (GRCm39, Ensembl release 106), respectively, with ERCC spike-in sequences included. Quality of the samples was assessed using FastQC (v0.11.9), Picard Tools CollectRnaSeqMetrics (v 2.26.0), and MultiQC (v1.11) (*Ewels et al., 2016*) to explore base quality, rRNA content, intronic percentages, and mapping rates. The Salmon transcript abundance estimates were aggregated to the gene level using tximport (v1.22.0) (*Soneson et al., 2015*). Additional quality control was performed on the gene-level counts using principal component analysis. Two samples were removed from the analysis due to a batch effect and a label mismatch. Differential expression analysis was performed with DESeq2 (v1.34.0) (*Love et al., 2014*), while over-representation analysis and gene set enrichment analysis (GSEA) of the differential expression results was executed using clusterProfiler (v4.2.2) (*Wu et al., 2021*) with GO (*Ashburner et al., 2000*; *Carbon et al., 2021*), KEGG (*Kanehisa, 2019*; *Kanehisa et al., 2023*; *Kanehisa and Goto, 2000*), and MsigDB (*Liberzon et al., 2015*; *Liberzon et al., 2011*; *Subramanian et al., 2005*) gene sets. GSEA was performed on the apeglm (v1.16.0) (*Zhu et al., 2019*)-shrunken log2 fold changes output from DESeq2.

## iSPAG7 KO study design

Cre/ERT2 heterozygous, SPAG7-floxed homozygous animal were evaluated alongside Cre/ERT2 WT, SPAG7-floxed homozygous littermates. At 7 weeks of age female animals were injected IP with 200 mg/kg of tamoxifen (Millipore Sigma, Burlington, MA) dissolved in corn oil for two consecutive days.

## Placenta histological analysis

Placenta from Het x Het breedings were collected at e18.5 and fixed in 10% formalin. Tissues were embedded in paraffin and cut into 5 μm sections. A section from each animal was stained with hematoxylin and eosin using an automatic slide stainer (Leica) and microscopic evaluation was performed by a veterinary pathologist. The junctional zone and whole placenta area, excluding residual decidua, were measured using FIJI software (2.9.0) blinded to the experimental groups. Labyrinth area was calculated as the whole placenta area – junctional zone area. The percentage of junctional zone present was calculated as the junctional zone area / whole placenta area (excluding decidua) x 100. The labyrinth:junctional zone ratio was calculated as the labyrinth area / junctional zone area.

## mtDNA copy number

Gastrocnemius muscle tissue was dissected from WT and SPAG7 KO animals. Tissues were pulverized and DNA was extracted using phenol:chloroform:isoamyl alcohol (25:24:1) followed by ethanol precipitation. mtDNA copy number was determined via RT-qPCR using a probe for Mitochondrial Cytochrome b (Fisher Scientific - Mm04225271_g1). Nuclear DNA copy number was determined using a probe for Ubiquitin C (Fisher Scientific - Mm02525934_g1). Data are expressed as a ratio of Mitochondrial Cytochrome b expression per Ubiquitin c expression.

## Statistical analysis

Statistical analysis, including post-hoc tests and statistical significance, were determined using Welch's two-way two sample t-test, one-way ANOVA, and Tukey HSD. Exact sample size (n) is included in

figure legends. Data are expressed as mean +/-standard deviation. Data were estimated to be statistically significant when p<0.05.

## Additional information

### Competing interests

Stephen E Flaherty, LouJin Song, Mary Piper, Shoh Asano, John D Griffin, Andrew Robertson, Dinesh Hirenallur Shanthappa, Youngwook Ahn, Evanthia Pashos: employee of Pfizer Inc. Olivier Bezy, Brianna LaCarubba Paulhus, Jincheng Pang, Yoson Park, Alan Opsahl, Morris J Birnbaum, Zhidan Wu: was an employee of Pfizer Inc when the study was conducted. The other authors declare that no competing interests exist.

### Funding

| Funder | Grant reference number | Author |
| --- | --- | --- |
| Pfizer Pharmaceuticals | Research and Development | Rebecca A Simmons |

The funders had no role in study design, data collection and interpretation, or the decision to submit the work for publication.

### Author contributions

Stephen E Flaherty III, Conceptualization, Data curation, Formal analysis, Investigation, Methodology, Writing – original draft, Writing – review and editing; Olivier Bezy, Conceptualization, Data curation, Formal analysis, Funding acquisition, Investigation, Writing – review and editing; Brianna LaCarubba Paulhus, Data curation, Investigation, Methodology, Writing – review and editing; LouJin Song, Mary Piper, Jincheng Pang, Shoh Asano, Yu-Chin Lien, John D Griffin, Andrew Robertson, Alan Opsahl, Dinesh Hirenallur Shanthappa, Youngwook Ahn, Evanthia Pashos, Rebecca A Simmons, Data curation, Formal analysis, Investigation, Methodology, Writing – review and editing; Yoson Park, Data curation, Formal analysis, Methodology, Writing – review and editing; Morris J Birnbaum, Conceptualization, Supervision, Investigation, Project administration, Writing – review and editing; Zhidan Wu, Conceptualization, Data curation, Formal analysis, Supervision, Investigation, Writing – original draft, Project administration, Writing – review and editing

### Author ORCIDs

Stephen E Flaherty III, https://orcid.org/0000-0002-5050-6451
LouJin Song https://orcid.org/0000-0002-1646-8121
Mary Piper https://orcid.org/0000-0003-2699-3840
Zhidan Wu https://orcid.org/0000-0001-7289-9420

### Ethics

All animal experiments were conducted following study protocols and procedures reviewed and approved by Pfizer Institutional Animal Care and Use Committee. The facilities that supported this work are fully accredited by AAALAC International.

Reviewer #1 (Public review): https://doi.org/10.7554/eLife.91114.3.sa1
Reviewer #2 (Public review): https://doi.org/10.7554/eLife.91114.3.sa2
Reviewer #3 (Public review): https://doi.org/10.7554/eLife.91114.3.sa3
Author response https://doi.org/10.7554/eLife.91114.3.sa4

## Additional files

### Supplementary files
• MDAR checklist

## Data availability

The data discussed in this publication have been deposited in NCBI's Gene Expression Omnibus and are accessible through GEO Series accession number GSE246428 (https://www.ncbi.nlm.nih.gov/geo/query/acc.cgi?acc=GSE246428).

The following dataset was generated:

| Author(s) | Year | Dataset title | Dataset URL | Database and Identifier |
|---|---|---|---|---|
| Flaherty S, Piper M | 2023 | SPAG7 deletion causes intrauterine growth restriction, resulting in adulthood obesity and metabolic dysfunction | https://www.ncbi.nlm.nih.gov/geo/query/acc.cgi?acc=GSE246428 | NCBI Gene Expression Omnibus, GSE246428 |

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
