## [Editor Report · eLife assessment]

This study combines molecular genetics and target validation to discover genes involved in obesity and determine their role. It was unanimously agreed that the work is **important** in terms of significance as it has conceptual and practical implications beyond metabolism, including embryonic and placental development. The strength of evidence is **convincing** from the use of their forward genetic screen in mice.

---

## [Referee Report · Reviewer #1 (Public review)]

Drawing on insights from preceding studies, the researchers pinpointed mutations within the spag7 gene that correlate with metabolic aberrations in mice. The precise function of spag7 has not been fully described yet, thereby the primary objective of this investigation is to unravel its pivotal role in the development of obesity and metabolic disease in mice. First, they generated a mice model lacking spag7 and observed that KO mice exhibited diminished birth size, which subsequently progressed to manifest obesity and impaired glucose tolerance upon reaching adulthood. This behaviour was primarily attributed to a reduction in energy expenditure. In fact, KO animals demonstrated compromised exercise endurance and muscle functionality, stemming from a deterioration in mitochondrial activity. Intriguingly, none of these effects was observed when using a tamoxifen-induced KO mouse model, implying that Spag7's influence is predominantly confined to the embryonic developmental phase. Explorations within placental tissue unveiled that mice afflicted by Spag7 deficiency experienced placental insufficiency, likely due to aberrant development of the placental junctional zone, a phenomenon that could impede optimal nutrient conveyance to the developing fetus. Overall, the authors assert that Spag7 emerges as a crucial determinant orchestrating accurate embryogenesis and subsequent energy balance in the later stages of life.

The study boasts several noteworthy strengths. Notably, it employs a combination of animal models and a thorough analysis of metabolic and exercise parameters, underscoring a meticulous approach. Furthermore, the investigation encompasses a comprehensive evaluation of fetal loss across distinct pregnancy stages, alongside a transcriptomic analysis of skeletal muscle, thereby imparting substantial value. Upon addressing the previously mentioned aspects, the study is poised to exert a substantial influence on the field, its significance reverberating significantly. The methodologies and data presented undoubtedly hold the potential to facilitate the community's deeper understanding of the ramifications stemming from disruptions during pregnancy, shedding light on their enduring impact on the metabolic well-being of subsequent generations.

---

## [Referee Report · Reviewer #2 (Public review)]

Summary:

The authors of this manuscript are interested in discovering and functionally characterizing genes that might cause obesity. To find such genes, they conducted a forward genetic screen in mice, selecting strains which displayed increased body weight and adiposity. They found a strain, with germ-line deficiency in the gene Spag7, which displayed significantly increased body weight, fat mass, and adipose depot sizes manifesting after the onset of adulthood (20 weeks). The mice also display decreased organ sizes, leading to decreased lean body mass. The increased adiposity was traced to decreased energy expenditure at both room temperature and thermoneutrality, correlating with decreased locomotor activity and muscle atrophy. Major metabolic abnormalities such as impaired glucose tolerance and insulin sensitivity also accompanied the phenotype. Unexpectedly, when the authors generated an inducible, whole body knockout mouse using a globally expressed Cre-ERT2 along with a globally floxed Spag7, and induced Spag7 knockout before the onset of obesity, none of the phenotypes seen in the original strain were recapitulated. The authors trace this discrepancy to the major effect of Spag7 being on placental development.

Strengths:

Strengths of the manuscript are its inherently unbiased approach, using a forward genetic screen to discover previously unknown genes linked to obesity phenotypes. Another strong aspect of the work was the generation of an independent, complementary, strain consisting of an inducible knockout model, in which the deficiency of the gene could be assessed in a more granular form. This approach enabled the discovery of Spag7 as a gene involved in the establishment of the mature placenta, which determines the metabolic fate of the offspring. Additional strengths include the extensive array of physiological parameters measured, which provided a deep understanding of the whole-body metabolic phenotype and pinpointed its likely origin to muscle energetic dysfunction.

Weaknesses:

Weaknesses that can be raised are the lack of molecular mechanistic understanding of the numerous phenotypic observations. For example, the specific role of Spag7 to promote placental development remains unclear. Also, the reason why placental developmental abnormalities lead to muscle dysfunction, and whether indeed the entire metabolic phenotype of the offspring can be attributed solely to decreased muscle energetics is not fully explored.

Overall, the authors achieved a remarkable success in identifying genes associated with development of obesity and metabolic disease, discovering the role of Spag7 in placental development, and highlighting the fundamental role of in-utero development in setting future metabolic state of the offspring.

Comments on revised version:

I have no further comments on my assessment of this interesting paper.

---

## [Referee Report · Reviewer #3 (Public review)]

Summary:

The manuscript by Flaherty III S.E. et al identified SPAG7 gene in their forward mutagenetic screening and created the germline knockout and inducible knockout mice. The authors reported that the SPAG7 germline knockout mice had lower birth weight likely due to intrauterine growth restriction and placental insufficiency. The SPAG7 KO mice later developed obesity phenotype as result of reduced energy expenditure. However, the inducible SPAG7 knockout mice had normal body weight and composition.

Strengths:

In this reviewer's opinion, this study has high significance in the field of metabolic research for the following reasons.

1. The authors' findings are significant in the field of obesity research, especially from the perspective of maternal-fetal medicine. The authors created and analyzed the SPAG7 KO mice and found that the KO mice had a "thrifty phenotype" and developed obesity.

2. SPAG7 gene function hasn't been thoroughly studied. The reported phenotype will fill the gap of knowledge.

Overall, the authors have presented their results in a clear and logically organized structure, clearly stated the key question to be addressed, used the appropriate methodology, produced significant and innovative main findings.

Comments on revised version:

The authors have satisfactorily addressed my previous concerns.

---

## [Author Response]

The following is the authors’ response to the original reviews.

**Reviewer #1:**
Drawing on insights from preceding studies, the researchers pinpointed mutations within the spag7 gene that correlate with metabolic aberrations in mice. The precise function of spag7 has not been fully described yet, thereby the primary objective of this investigation is to unravel its pivotal role in the development of obesity and metabolic disease in mice. First, they generated a mice model lacking spag7 and observed that KO mice exhibited diminished birth size, which subsequently progressed to manifest obesity and impaired glucose tolerance upon reaching adulthood. This behaviour was primarily attributed to a reduction in energy expenditure. In fact, KO animals demonstrated compromised exercise endurance and muscle functionality, stemming from a deterioration in mitochondrial activity. Intriguingly, none of these effects was observed when using a tamoxifen-induced KO mouse model, implying that Spag7's influence is predominantly confined to the embryonic developmental phase. Explorations within placental tissue unveiled that mice afflicted by Spag7 deficiency experienced placental insufficiency, likely due to aberrant development of the placental junctional zone, a phenomenon that could impede optimal nutrient conveyance to the developing fetus. Overall, the authors assert that Spag7 emerges as a crucial determinant orchestrating accurate embryogenesis and subsequent energy balance in the later stages of life.The study boasts several noteworthy strengths. Notably, it employs a combination of animal models and a thorough analysis of metabolic and exercise parameters, underscoring a meticulous approach. Furthermore, the investigation encompasses a comprehensive evaluation of fetal loss across distinct pregnancy stages, alongside a transcriptomic analysis of skeletal muscle, thereby imparting substantial value. However, a pivotal weakness of the study centres on its translational applicability. While the authors claim that "SPAG7 is well-conserved with 97% of the amino acid sequence being identical in humans and mice", the precise role of spag7 in the human context remains enigmatic. This limitation hampers a direct extrapolation of findings to human scenarios. Additionally, the study's elucidation of the molecular underpinnings behind the spag7-mediated anomalous development of the placental junction zone remains incomplete. Finally, the hypothesis positing a reduction in nutrient availability to the fetus, though intriguing, requires further substantiation, leaving an aspect of the mechanism unexplored.Hence, in order to fortify the solidity of their conclusions, these concerns necessitate meticulous attention and resolution in the forthcoming version of the manuscript. Upon the comprehensive addressing of these aspects, the study is poised to exert a substantial influence on the field, its significance reverberating significantly. The methodologies and data presented undoubtedly hold the potential to facilitate the community's deeper understanding of the ramifications stemming from disruptions during pregnancy, shedding light on their enduring impact on the metabolic well-being of subsequent generations.

Thanks to this reviewer for their thoughtful analysis and commentary. Human mutations in SPAG7 are exceedingly rare (SPAG7 | pLoF (genebass.org)), potentially because of the deleterious effects of SPAG7-deficiency on prenatal development. This makes investigation into the causative effects of SPAG7 in humans challenging. There exist mutations in the SPAG7 region of the genome that are associated with BMI, but no direct coding variants within the spag7 gene itself have been studied.

We agree with the reviewer that the precise role of spag7 in the placenta remains unknown. However, given its robust expression and high protein levels in the placenta, including in key cells, such as the syncytiotrophoblast (https://www.proteinatlas.org/ENSG00000091640-SPAG7/tissue/Placenta), it is highly likely that spag7 is critical for normal placenta development and function. Multiple studies (https://www.ncbi.nlm.nih.gov/pmc/articles/PMC9716072/) have recently shown that sperm associated RNAs play a critical role in embryonic and early placenta development. Our findings will provide the basis for future studies that can elucidate the role of spag7 in human placenta.

**Reviewer #2:**
Summary:The authors of this manuscript are interested in discovering and functionally characterizing genes that might cause obesity. To find such genes, they conducted a forward genetic screen in mice, selecting strains which displayed increased body weight and adiposity. They found a strain, with germ-line deficiency in the gene Spag7, which displayed significantly increased body weight, fat mass, and adipose depot sizes manifesting after the onset of adulthood (20 weeks). The mice also display decreased organ sizes, leading to decreased lean body mass. The increased adiposity was traced to decreased energy expenditure at both room temperature and thermoneutrality, correlating with decreased locomotor activity and muscle atrophy. Major metabolic abnormalities such as impaired glucose tolerance and insulin sensitivity also accompanied the phenotype. Unexpectedly, when the authors generated an inducible, whole body knockout mouse using a globally expressed Cre-ERT2 along with a globally floxed Spag7, and induced Spag7 knockout before the onset of obesity, none of the phenotypes seen in the original strain were recapitulated. The authors trace this discrepancy to the major effect of Spag7 being on placental development.Strengths:Strengths of the manuscript are its inherently unbiased approach, using a forward genetic screen to discover previously unknown genes linked to obesity phenotypes. Another strong aspect of the work was the generation of an independent, complementary, strain consisting of an inducible knockout model, in which the deficiency of the gene could be assessed in a more granular form. This approach enabled the discovery of Spag7 as a gene involved in the establishment of the mature placenta, which determines the metabolic fate of the offspring. Additional strengths include the extensive array of physiological parameters measured, which provided a deep understanding of the whole-body metabolic phenotype and pinpointed its likely origin to muscle energetic dysfunction.Weaknesses:Weaknesses that can be raised are the lack of molecular mechanistic understanding of the numerous phenotypic observations. For example, the specific role of Spag7 to promote placental development remains unclear. Also, the reason why placental developmental abnormalities lead to muscle dysfunction, and whether indeed the entire metabolic phenotype of the offspring can be attributed solely to decreased muscle energetics is not fully explored.Overall, the authors achieved a remarkable success in identifying genes associated with development of obesity and metabolic disease, discovering the role of Spag7 in placental development, and highlighting the fundamental role of in-utero development in setting future metabolic state of the offspring.

We thank this reviewer for their thoughtful analysis and commentary. Significant effort has been made to understand the causes of the metabolic phenotypes observed in SPAG7-deficient mouse models. It is clear that hyperphagia is not the cause and the muscle energetics deficit is likely not the sole cause. We expect that decreased access to nutrition in utero will lead to widespread and varied metabolic adaptation.

We agree with the reviewer that further work can be done to understand the molecular mechanism driving the metabolic phenotypes of SPAG7-deficient animals. We believe that full investigation of the processes behind the developmental abnormalities is beyond the scope of this paper and best to be done under a separate paper.

**Reviewer #3:**
Summary:The manuscript by Flaherty III S.E. et al identified SPAG7 gene in their forward mutagenetic screening and created the germline knockout and inducible knockout mice. The authors reported that the SPAG7 germline knockout mice had lower birth weight likely due to intrauterine growth restriction and placental insufficiency. The SPAG7 KO mice later developed obesity phenotype as a result of reduced energy expenditure. However, the inducible SPAG7 knockout mice had normal body weight and composition.Strengths:In this reviewer's opinion, this study has high significance in the field of metabolic research for the following reasons.1. The authors' findings are significant in the field of obesity research, especially from the perspective of maternal-fetal medicine. The authors created and analyzed the SPAG7 KO mice and found that the KO mice had a "thrifty phenotype" and developed obesity.1. SPAG7 gene function hasn't been thoroughly studied. The reported phenotype will fill the gap of knowledge.Overall, the authors have presented their results in a clear and logically organized structure, clearly stated the key question to be addressed, used the appropriate methodology, produced significant and innovative main findings.Weaknesses:The manuscript can be further strengthened with more clarification on the following points.1. The germline whole-body KO mice were female mice (Line293), however the inducible knockout mice were male mice (Line549). Sexual dimorphism is often observed in metabolic studies, therefore the metabolic phenotype of both female and male mice needs to be reported for the germline and inducible knockouts in order to make the justified conclusion.1. SPAG7 has an NLS. Does this protein function in gene expression? Whether the overall metabolic phenotype is the direct cause of SPAG7 ablation is unclear. For example, the Hsd17b10 gene was downregulated in all tissues in the KO mice. Could this have been coincidentally selected for and thus be the cause of the developmental issues and adulthood obesity? Do the iSpag7 mice demonstrate reduced expression of Hsd17b10?1. Figure 2c should display the energy expenditure normalized to body weight (or lean body mass).1. Please provide more information for the figure legend, including the statistical test that was conducted for each data set, animal numbers for each genotype and sexes.1. The authors should report how long after treatment the data was collected for figures 4F-M.1. The authors should justify ending the data collection after 8 weeks for the iSPAG7 mice in Figures 4C-E. In the WT vs germline KO mice, there was no clear difference in body weight or lean mass at 15 weeks of age.

Response to point #1 (Weakness): We thank the reviewer for their thoughtful analysis and commentary. All inducible KO animals described in the paper are female (the typo in Line 549 has been corrected). We did perform studies in both male and female animals for both of these lines. Males display similar metabolic phenotypes, though not as robustly as the females. A table summarizing key data from male and female germline KO animals and inducible KO animals has been included below.

**Author response table 1. sa4table1:** 

	SPAG7 KO			
Body Weight Week 6(g)	18.76+-0.84	18.00+-0.33	23.36+-0.74	22.47+-1.81
Lean Mass Week 6(g)	16.76+-0.99	15.04+-0.29	21.18+-0.34	19.55+-1.32
Fat Mass Week 6(g)	1.54+-0.27	2.53+-0.48	1.47+-0.24	2.42+-0.56
Body Weight Week 20(g)	22.53+-1.70	22.31+-1.72	28.6+-2.95	30.47+-3.18
Lean Mass Week 20(g)	21.12+-1.04	19.78+-1.04	26.25+-1.40	24.42+-1.14
Fat Mass Week 20(g)	3.43+-1.20	8.50+-3.87	4.15+-1.69	6.72+-1.29
Food Intake Week 20(g)	3.32+-0.77	3.01+-0.74	3.84+-0.62	3.87+-1.00
Total Energy Expenditure Week 11 (kcal)	31.80+-1.37	27.75+-2.76	32.02+-1.49	26.94+-3.63
oGTT AUC Week 12	1227+-94.8	1761+-110	1139+-92	1971+-127

**Author response table 2. sa4table2:** 

	F WT	F HOM	M WT	M HOM
Body Weight Week 0 (g)	19.86+-2.84	19.55+-1.31	25.06+-2.84	24.4+-0.75
Body Weight Week 9 (g)	22.53+-1.70	22.31+-1.72	28.6+-2.95	30.47+-3.18
Lean Mass Week 9 (g)	19.61+-0.85	19.37+-1.02	25.07+-2.69	25.42+-1.99
Fat Mass Week 9 (g)	3.31+-0.79	2.85+-0.92	3.53+-1.23	3.70+-0.80
Food Intake Week 9 (g)	2.12+-0.56	2.08+-0.30	2.20+-0.44	2.18+-0.38
Total Energy Expenditure Week 6 (kcal)	29.71+-1.28	30.72+-1.43	31.13+-1.26	31.25+-1.2
oGTT AUC Week 7	3503+-178	3601+-180	3217+-145	3306+-151

Response to point #2 (Weakness): SPAG7 contains an R3H domain, which is predicted to bind polynucleotides, and other proteins that contain R3H domains are known to bind RNA or ssDNA. The iSPAG7 mice do display decreased hsd17b10 expression (to a lesser degree than the germline KOs) in the tissues examined. When we knock-down SPAG7 in specific tissues, we also see hsd17b10 expression decrease specifically in those tissues. These data all suggest that hsd17b10 expression is, at least, linked to spag7 expression. They also raise the question of why these animals have no metabolic phenotype. Some possible explanations are that hsd17b10 expression is essential only during early development, or that the lower magnitude of downregulation of hsd17b10 in the iSPAG7 is insufficient to produce the metabolic phenotypes seen in the germline Kos with higher magnitude of downregulation.

Response to point #3 (Weakness): How best to normalize total energy expenditure data is a subject of debate within the energy expenditure field. As the animals have increased body weight and decreased lean mass, normalizing to either will skew the results in different directions. We have included the data normalized to body weight and to lean mass below. The decrease in total energy expenditure remains significant in either scenario.

**Author response image 1. sa4fig1:** 

Response to point #4 (Weakness): The information has been added to all figures.

Response to point #5 (Weakness): Weeks after treatment have been added to the figure legends for Figures 4F-M.

Response to point #6 (Weakness): Highly significant changes in fat mass, glucose tolerance and insulin sensitivity are already present in the germline SPAG7 KO mice at age of 15 week or earlier. Tamoxifen injection effectively induced SPA7 gene KO in less than a week in the iSPAG7 KO mice. Given the absence of significant changes or any trends towards significance in glucose and insulin tolerance test as well as other metabolic testes in the iSPAG7 KO mice at age of 15 week (same age as the germline KO when these changes observed) and 8 week after SPAG7 gene KO, we did not anticipate to see the changes beyond this point and decided to stop the study at 9 weeks after treatment.